# Correlating Grip Force Signals from Multiple Sensors Highlights Prehensile Control Strategies in a Complex Task-User System

**DOI:** 10.3390/bioengineering7040143

**Published:** 2020-11-10

**Authors:** Birgitta Dresp-Langley, Florent Nageotte, Philippe Zanne, Michel de Mathelin

**Affiliations:** 1ICube UMR 7357, Centre National de la Recherche Scientifique (CNRS), 75016 Paris, France; 2ICube UMR 7357 Robotics Department, University of Strasbourg, 67081 Strasbourg, France; nageotte@unistra.fr (F.N.); philippe.zanne@unistra.fr (P.Z.); demathelin@unistra.fr (M.d.M.)

**Keywords:** wearable biosensors, wireless technology, human grip force, motor control, complex task-user systems, expertise, multivariate data, correlation analysis, functional analysis

## Abstract

Wearable sensor systems with transmitting capabilities are currently employed for the biometric screening of exercise activities and other performance data. Such technology is generally wireless and enables the non-invasive monitoring of signals to track and trace user behaviors in real time. Examples include signals relative to hand and finger movements or force control reflected by individual grip force data. As will be shown here, these signals directly translate into task, skill, and hand-specific (dominant versus non-dominant hand) grip force profiles for different measurement loci in the fingers and palm of the hand. The present study draws from thousands of such sensor data recorded from multiple spatial locations. The individual grip force profiles of a highly proficient left-hander (expert), a right-handed dominant-hand-trained user, and a right-handed novice performing an image-guided, robot-assisted precision task with the dominant or the non-dominant hand are analyzed. The step-by-step statistical approach follows Tukey’s “detective work” principle, guided by explicit functional assumptions relating to somatosensory receptive field organization in the human brain. Correlation analyses (Person’s product moment) reveal skill-specific differences in co-variation patterns in the individual grip force profiles. These can be functionally mapped to from-global-to-local coding principles in the brain networks that govern grip force control and its optimization with a specific task expertise. Implications for the real-time monitoring of grip forces and performance training in complex task-user systems are brought forward.

## 1. Introduction

Wireless sensor systems with transmitting capabilities represent innovative technology that can be exploited to monitor specific physiological activities related to task performance [1]. The latest wireless devices allow for unobtrusive monitoring of signals in real time. Such signals include signals relative to individual grip force data recorded from multiple sensor locations in a user’s hands. Recent sensor designs, electronics, data/power management, and signal processing capabilities permit the statistical analysis of thousands of such data (big data) in minimal computation time. However, many technological aspects still remain to be optimized. For example, to read functional meaning into such a multitude of signal variations, hypothesis-driven methods of analysis and visualization are needed, as will be shown here on the example of step-by-step statistical “detective work” inspired by functional hypotheses and theory. John W. Tukey [2] was among the first to explicitly compare the analysis of complex data to “good detective work”. Such should not be fully automated, but may benefit from wisely and adequately developed automated steps within a functionally inspired analytical strategy. Good detective work in multivariate data analysis has to start by asking the right question(s), then look for the right indicators (clues) and, finally, draw the right conclusions from the clues made available. This is illustrated here on the example of thousands of grip force signals recorded from multiple locations in the dominant and dominant hands of users with varying levels of task expertise, i.e., proficiency in the manual control of a robotic precision task device. How functionally meaningful signal correlations are obtained from a multitude of spatiotemporally variable grip force profiles is illustrated. 

Human grip force is controlled at several hierarchical stages, from sensory receptors to the brain, and back to the hand. Its functional aspects have been relatively well studied. For example, the relationship between individual finger positions and grip forces of men and women were studied on subjects holding a cylindrical object in a circular precision grip in the five-, four-, three-, and two-finger grip modes [3]. When the individual finger position is constant for all weights and diameters, with no differences in individual finger position, gender specific hand dimension, or strength, total grip force increases as the number of fingers used for grasping decreases [3]. In the contribution of the individual fingers to total grip force, the contribution of the thumb is followed by that of the ring and the little finger. The contribution of the index finger is the smallest, and there is no gender difference for any of the grip force variables tested [3]. The effects of hand dimension and hand strength on the individual finger grip forces are generally subtle and minor. The contribution and co-ordination of finger grip forces in precision grip tasks using multiple fingers is reflected by large changes in finger force recorded from the index finger, followed by the middle, ring, and little fingers [4]. Results suggest that all individual finger force adjustments for loads less than 800 g are controlled by a single common scaling value; the fewer the number of fingers used, the greater the total grip force [4]. The grip mode, i.e., whether all five or only three fingers are used, influences the force contributions of the middle and ring fingers, but not that of the index finger [4]. Other studies have shown digit redundancy in human prehensile behavior [5]. The partly redundant design of the hand allows performing a variety of tasks in a reliable and flexible way following the principle of abundance, as shown in robotics with respect to the control of artificial grippers, for example. Multi-digit synergies appear to operate at two levels of hierarchy to control prehensile action [5]. Forces produced by the thumb and the “virtual finger” (an imagined finger with a mechanical action equal to the combined mechanical action of all other four fingers of the hand) co-vary at a higher level only to stabilize the grip action in respect to the orientation of the hand-held object. Analysis of grip force adjustments during motion of hand-held objects suggests that the central nervous system adjusts to gravitational and inertial loads differently, at an even higher level of control. Object manipulation by efficient control of finger and grip force is therefore not only a motor skill, but also a highly cognitive skill [6] that requires multisensory integration and is exploited in surgery, craft making, and musical performance. Sequences of relatively straightforward cause-effect links directly related to mechanical constraints lead on to a non-trivial co-variation between low-level and high-level control variables [4,5,6], as in playing a musical instrument, which requires independent control of the magnitude and rate of force production, which typically vary in relation to loudness and tempo [6]. The expert performance of a highly skilled pianist, for example, is characterized by a rapid reduction of finger forces, allowing for considerably fast performance of repetitive piano keystrokes [6]. Skilled grasping behavior (multi-finger grasping) has three essential components: (1) manipulation force, or resultant force and moment of force exerted on the object and the digits’ contribution to force production, (2) internal forces, which are defined as forces that cancel each other out to maintain object stability, to ensure slip prevention, tilt prevention, and robustness against perturbation, and (3) motor control (or grasp control), which involves prehensile synergies, chain effects, inter-finger connection, and the high-level brain command of simultaneous digit adjustments to several mutually reinforcing or conflicting demands [3,4,5,7,8]. Prehensile synergies are reflected by the characteristics of single digit actions and their co-variation patterns during task execution [7,8]. Sharing of force patterns across the digits minimizes mechanically unnecessary digit forces, resulting in a trade-off between multi-digit synergies at the two levels of control hierarchy mentioned here above [9]. The force contributions of the index and the middle finger are generally the most important [10], followed by combinations of the ring and little finger (pinky). The dominant hand in general generates smaller force contributions of the thumb and the ring finger, and greater force contributions of the palm of the hand [10]. Left handers are less consistent compared to right handers in performing better with their dominant hands [11]. It is currently suggested that the right and left brain hemispheres may each represent the movements of the contralateral, not the ipsilateral hand, however, the programming of isometric fingertip forces is initially based on internal memory representations of physical device properties in addition to visual and haptic information from the current manipulations. These central memory representations are not necessarily lateralized [12,13,14,15,16], and permit anticipatory control of forces, scaled in advance, as explained here above. This allows for a quick and accurate adjustment of grip forces well beyond limitations due to lateralization of internal representation, or a sole dependency on ongoing sensory feedback. 

The functionally specific neural networks for grip force representation and control relate to the somatosensory cortical network [16], or the so-called S1 map, in the primate brain. S1 refers to a neocortical area that responds primarily to tactile stimulations on the skin or hair. Somatosensory neurons have the smallest receptive fields and receive the shortest-latency input from the receptor periphery. The S1 cortical area is conceptualized in current state of the art [16,17] as containing a single map of the receptor periphery. The somatosensory cortical network has a modular functional architecture and connectivity, with highly specific connectivity patterns [16,17,18,19,20], binding functionally distinct neuronal subpopulations from other cortical areas into motor circuit modules at several hierarchical levels [17]. The functional modules display a hierarchy of interleaved circuits connecting via inter-neurons in the spinal cord, in visual sensory areas, and in motor cortex with feed-back loops, and bilateral communication with supraspinal centers [17,18]. The from-local-to-global functional organization (Figure 1) of the somatosensory neural circuits relates to precise connectivity patterns, and these patterns frequently correlate with specific behavioral functions or motor output. Current state of the art suggests that developmental specification, where neuronal subpopulations are specified in the process of a precisely timed neurogenesis [17,18], determines the self-organizing nature of this connectivity for motor control and, in particular, limb movement control [20]. The functional plasticity of the somatosensory cortical network is revealed by neuroscience research on the human primate [19] showing that somatosensory representations of fingers left intact after amputation of others on the same hand become expanded in less than ten days after amputation, when compared with representations in the intact hand of the same patient, or to representations in either hand of controls. Such a network expansion reflects the functional resiliency of the self-organized somatosensory system, which has evolved as a function of active constraints [20], and in harmony with other sensory systems such as the visual and auditory brain. Grip force profiles are a direct reflection of the complex low-level, cognitive, and behavioral synergies this evolution has produced. The anterior intraparietal area (AIP) of the somatosensory map in particular seems involved in the internal processing and control of precision grasping, regardless of the hand in use [14,15], indicating that central grip force representation in the brain is not lateralized despite observable differences between forces produced by the dominant and non-dominant hands. Current neuroanatomy of the brain circuitry that controls individual digit movements and grip forces generated during such movements displays a from-global-to-local functional organization. Neuroimaging studies have shown representations in motor cortex (M1) of individual digit movements, resulting either in largely overlapping neural activity patterns with specific voxels responding to all moving fingers at the same time [21], or in a strictly somatotopic organization (functional map) with locally ordered representations of individual finger digit movements (flexion), from thumb to pinky [22].

Given that internal grip force representation appears both locally mapped and globally encoded in the neural networks of the somatosensory brain, the question addressed here in this study is whether variations in skill or expertise may result in distinctive from-global-to-local grip force co-variation patterns across fingers and hands. While there is digit redundancy for spontaneous grip force production in simple grasping tasks where no particular skill is required [5], such redundancy may be minimized in the skillful performance of a highly demanding precision task. In other words, while the complete novice deploys digit forces nonspecifically for lack of skill in a precision grip task, the skilled performance of the expert should display optimal correlation of specific finger forces. This assumption motivated the step-by-step analyses provided here below, aimed at highlighting local grip force co-variation patterns characteristic of the prehensile control strategy of an expert. In particular, it will be shown that the statistical correlation patterns between locally produced grip forces in the individual profiles of users with different levels of training deliver insight into the ways in which functional redundancies are minimized with training. This is shown and discussed in the context of a complex task-device-user system with tightly limited degrees of freedom for hand and finger movement. 

## 2. Materials and Methods

A wireless wearable (glove) sensor system was used for collecting thousands of grip force data per sensor location and individual user in real time. The glove system does not provide haptic feedback to the user. The robotic system is designed for bi-manual intervention, and task simulations may solicit either the dominant or the non-dominant hand, as in our case here, or both hands at the same time, depending on the complexity of interventions. Here, four successive steps of a pick-and-drop task requiring execution with either the dominant or the non-dominant hand through manipulation of the corresponding grip handle instrument of the system are exploited.

### 2.1. Slave Robotic System 

The slave robotic system is built on the Anubis^®^ platform of Karl Storz and consists of three flexible, cable-driven sub-systems for robot-assisted endoscopic surgery, described in detail in our previous work [23,24,25,26]. A recent description can be found, for example, in the open access publication [24]. The most essential of the task-user system specifics are described again here, in similar wording, for the readers’ convenience: The endoscope has two lateral channels which are deviated from the main direction by two flaps at the distal extremity. The instruments have bending extremities (in one direction) and can be inserted inside the channels of the endoscope. The system has a tree-like architecture and the movements of the endoscope act also upon the position and orientation of the electrical and mechanical instruments. Overall, the slave system has 10 motorized degrees of freedom (DoF). The main endoscope carries a fisheye camera at its tip, providing visual feedback, and can be bent in two orthogonal directions. This allows moving the endoscopic view from left to right, from up to down, and from forward to backward. Each instrument has three DoF: translation (*t**z*) and rotation (*θ**z*) in the endoscope channel, and deflection of the active extremity (angle *β*). The deflection is actuated by cables running through the instrument body from the proximal part up to the distal end. The mechanical instruments can be opened and closed. 

### 2.2. Master/Slave Control 

The master side consists of two specially designed interfaces, which are passive mobile mechanical systems. The user grasps two handles, each having 3 DoF, which translate for controlling instrument insertion, rotate around a horizontal axis for controlling instrument rotation, and rotate around a final axis (moving with the previous DoF) for controlling instrument bending. These DoFs are similar to the possible motions of the instruments as demonstrated in preclinical trials [23]. The robot controller runs on a computer (DELL Precision T5810 model computer equipped with an Intel Xeon CPU E5-1620 with 16 Giga bytes memory (RAM) under real-time Linux, which communicates with the master interfaces and the slave robot. The positions of the joints of the master interfaces are obtained from encoders, which are read at 1 kHz by the controller. The controller maps these positions individually to the corresponding joints of the slave instrument as reference positions. Mapping scales from master to slave are 1:1 for rotation, 1:2 for bending, and 1.2:1 for translation. The controller sends the joints’ references to the drivers controlling the slave motors at 1 kHz. The slave joints are individually served to their reference positions by their drivers. For additional information on the technical aspects of the control system, we refer the reader to [26]. Each handle is also equipped with a trigger and with a small four-way joystick for controlling camera movements. In the experiments here, the trigger is operated with the index finger of a given hand for controlling grasper opening and closing, and the small joysticks for moving the endoscope are not used during task execution. 

The user sits in front of the master console and looks at the endoscopic camera view displayed on the screen in front of him/her at a distance of about 80 cm while holding the two master handles, which are about 50 cm away from each other. Seat and screen heights are adjustable to optimal individual comfort. The two master interfaces are identical and the two slave instruments they control are also identical. Therefore, for a given task the same movements need to be produced by the user whatever the hand he/she uses (left or right). The master interfaces are statically balanced and all joints exhibit low friction, and therefore only minimal forces are required to produce movements in any direction. A snapshot view of a user wearing the sensor gloves while manipulating the handles of the system is shown in Figure 2a here above. The master-slave control chart of the master/slave system is displayed in Figure 2b. Figure 2c shows the different directions and types of tool-tip and control movements.

### 2.3. Sensor Glove Design 

The system having its own grip style design, a specific wearable sensor system in terms of two gloves, one for each hand, with inbuilt force sensitive resistors (FSR) was developed. The hardware and software configurations are described here below.

#### 2.3.1. Hardware 

The gloves designed for the study contain 12 FSR, in contact with specific locations on the inner surface of the hand as illustrated in Figure 3. Two layers of cloth were used and the FSR were inserted between the layers. The FSR did not interact, neither directly with the skin of the subject, nor with the master handles, which provided a comfortable feel when manipulating the system. FSR were sewn into the glove with a needle and thread. Each FSR was sewn to the cloth around the conducting surfaces (active areas). The electrical connections of the sensors were individually routed to the dorsal side of the hand and brought to a soft ribbon cable, connected to a small and very light electrical casing, strapped onto the upper part of the forearm and equipped with an Arduino microcontroller. Eight of the FSR, positioned in the palm of the hand and on the finger tips, had a 10 mm diameter, while the remaining four located on middle phalanxes had a 5 mm diameter. Each FSR was soldered to 10 KΩ pull-down resistors to create a voltage divider, and the voltage read by the analog input of the Arduino is given by
*V_out_* = *R_PD_V*_3.3_/(*R_PD_*+*R_FSR_*)(1)
where *R_PD_* is the resistance of the pull-down resistor, *R_FSR_* is the FSR resistance, and *V*_3.3_ is the 3.3 V supply voltage. FSR resistances can vary from 250 Ω when subject to 20 Newton (N) to more than 10 MΩ when no force is applied at all. The generated voltage varies monotonically between 0 and 3.22 Volt, as a function of the force applied, which is assumed uniform on the sensor surface. In the experiments here, forces applied did not exceed 10N, and voltages varied within the range of [0; 1500] mV. The relation between force and voltage is almost linear within this range. It was ensured that all sensors provided similar calibration curves. Thus, all following comparisons are directly between voltage levels at the millivolt (mV) scale. A regulated 3.3 V was provided to the sensors from the Arduino. Power was provided by a 4.2 V Li-Po battery enabling use of the glove system without any cable connections. The battery voltage level was controlled during the whole duration of the experiments by the Arduino and displayed continuously via the user interface. The glove system was connected to a computer for data storage via Bluetooth enabled wireless communication at a rate of 115,200 bits-per-second (bps).

#### 2.3.2. Software

The software of the glove system was divided into two parts: one running on the gloves, and one for data collection. The general design of the glove system is described as follows. Each of the two gloves was sending data to the computer separately, and the software read the input values and stored them on the computer according to their header values indicating their origin. The software running on the Arduino was designed to acquire analog voltages provided by the FSR every 20 milliseconds (50 Hz). In every loop, input voltages were merged with their time stamps and sensor identification. This data package was sent to the computer via Bluetooth, which was decoded by the computer software. The voltage data were saved in a text file for each sensor, with their time stamps and identifications. Furthermore, the computer software monitored the voltage values received from the gloves via a user interface showing the battery level. In case the battery level drops below 3.7 V, the system warns the user to change or charge the battery. However, this never occurred during the experiments reported here. 

### 2.4. Experimental Precision Grip Task

The experimental precision grip task is a 4-step pick-and-drop with robot-assistance. For visual illustration of device movements and a verbal description of each step, see Figure 4 below. During the experiments, only one of the two instruments controlling the tool-tips (left or right, depending on the task session) was moved, while the main endoscope and its image remained still. The experiments started with the right- or left-hand gripper being pulled back. Then the user had to approach the object (step 1) with the distal tool extremity by manipulating the handles of the master system effectively. Then, the object had to be grasped with the tool (step 2). Once firmly held by the gripper, the object had to be moved to a position on top of the target box (step 3) with the distal extremity of the tool in the correct position for dropping the object into the target box without missing (step 4). To drop the object, the user had to open the gripper of the tool. The user started and ended a given task session by pushing a button, wirelessly connected to the computer handling the data storage. 

### 2.5. Definition of Task Expertise

The robot-assisted precision grip task described here above only involves positioning the instrument tip through movements from left to right, from up to down, and from forward to backwards. Manipulation of one instrument only of is required to perform this task here. Therefore, users inevitably have limited degrees of freedom for hand and finger movements, and are bound to perform the task in rather the same way generally speaking. The locally deployed grip forces will vary depending on how skilled a user has become in performing the task accurately. Here, grip force data were collected from three users with distinctly different levels of task expertise. One can be considered an expert who had been practicing on the system since its manufacturing and who is currently the most proficient user, with years of user experience and more than 100 h of training in the specific task exploited here in this study. The expert is familiar with using the system with his dominant and his non-dominant hand. Another user had more than 50 h of task-specific training with the right dominant hand and can therefore be considered a dominant-hand-trained user. The third user was a complete novice who was given one hour to familiarize himself with the system prior to the experiment, and had never used the system before, nor had he any prior experience with any similar system. The three users’ hand sizes were about the same, and the sensor gloves were developed specifically to fit the hands of average-sized male individuals. The expert was left handed while the trained user and the novice were both right handed. Several thousands of grip force data were collected from the twelve sensor locations in the dominant and non-dominant hands of each of the three users in ten or more successive task sessions with each hand. Their distinct levels of expertise are consistently reflected by performance task parameters such as average task session times, or the number of task incidents in terms of of object drops, misses, and tool-trajectory adjustments during individual performance across a set of ten successive sessions; these individual expertise-specific descriptors are summarized here below in Table 1. As explained earlier, the left and right interfaces are identical, and the same task is realized with either hand.

## 3. Results

A total of 459,348 grip force signals was recorded in the experiment, corresponding to a total of 38,279 grip force signals per sensor. For the expert user, we have a total of 4442 data per sensor from the dominant left hand recorded in real time across ten successive task sessions, and 6116 data per sensor from the non-dominant right hand recorded in real time across ten successive task sessions. For the dominant-hand-trained user, we have a total of 5974 data from the dominant right hand across ten task sessions, and 6760 data from the non-dominant left hand across ten task sessions. For the total beginner (novice), we have 8484 data from the dominant right hand across eleven sessions, and 6496 data from the non-dominant left hand across eleven sessions. The full dataset is provided in Appendix A.

### 3.1. Data Preprocessing

All data analyses were performed on the original raw data from the ten or eleven repeated task sessions of each individual. A grip force signal from each sensor was acquired every 20 milliseconds in task time across sessions, therefore, a larger number of data from a given individual reflects longer task times. In the specific task here, different finger and hand locations are variably solicited during task execution. Significant differences as a function of task expertise can be expected, as shown in our previous work [21,22]. Here, prior to further statistical analysis, a descriptive evaluation was performed to establish which of the twelve sensors produced task-relevant output data in terms of signals of at least 50 mV in at least 10% of the data. The study goal here is to reveal functionally specific and potentially skill-dependent co-variation patterns in the finger forces of the expert by comparison with the non-experts. All further statistical analyses towards this goal, such as analysis of variance on the raw data (general linear model), or between-sensor correlation statistics, require comparing between signal distributions with roughly the same amount of data and variance in the data. This holds for all comparisons that follow here after elimination of sensor locations 1 and 4. Outliers in terms of values exceeding twice the amount of variance were not identified in any of the given distributions.

### 3.2. Descriptive Analyses

The column data from the spreadsheets were submitted to computation of the column means and the variance around the means in terms of standard errors. The results from these computations on the total number recorded for each sensor across users, hands, and sessions are shown here below in Figure 5. This first analysis reveals that the sensor locations S1 and S4 on the distal phalanx of the thumb and the middle phalanx of the index were not sufficiently solicited during task execution to produce a signal in at least 10% of the total data recorded. As a consequence, S1 and S4 were eliminated from subsequent analyses. Further detailed analyses for each sensor, user, and hand, shown here below in Figure 6a–c for the remaining ten sensors, for each user and hand. These reveal insignificant activity levels in sensors 1–4, 8, and 11 in the dominant hand of the expert (Figure 6a, left), and in sensors 1, 3, 4, 9, and 11 in the non-dominant hand (Figure 6a, right). The dominant hand of the trained user produced insignificant activity levels in sensors 1–4, 7, and 11 (Figure 6b, left), and the non-dominant hand produced insignificant activity levels in sensors 1, 3–5, 9, 11, and 12 (Figure 6b, right). In the novice data, insignificant activity levels in sensors 1 and 4 in the dominant hand (Figure 6c, left), and in sensors 1, 3, 4, 9, and 11 in the non-dominant hand (Figure 6c, right) were found.

Significant activity levels were recorded from sensor locations 5, 6, 7, 10, and 12 in the dominant hand of the expert (Figure 6a, left), and in sensors 5–11 and 12 in the non-dominant hand (Figure 6a, right). The dominant hand of the trained user produced significant task-relevant activity levels in sensors 5–10 and 12 (Figure 6b, left), and the non-dominant hand produced significant activity levels in sensors 2, 6–8, and 10 (Figure 6b, right). In the novice data, significant activity levels are found all sensors except 1 and 4 in the dominant hand (Figure 6c, left), and in sensors 5–8, 10, and 12 in the non-dominant hand (Figure 6c, right). These descriptive analyses reveal, as a common characteristic to all users and hands, non-significant activation of sensors 1 and 4, and a considerable variation in significant task-relevant sensor activities between users and hands. As a consequence, a two-way analysis of variance (ANOVA) comparing between the three levels of the ‘user’ factor (expert, trained, novice) and the two levels of the ‘hand’ factor (dominant, non-dominant) to assess the statistical significance of between-factor variations for each sensor in which task-relevant activity was found in at least one user and hand, i.e., S2, S3, S5, S6, S7, S8, S9, S10, S11, and S12.

### 3.3. Analysis of Variance

Further analyses of variance (two-way ANOVA) were run separately for each of the active sensors as defined here above. The analyses were performed using *SYSTAT* with *SIGMAPLOT_12*, and returned statistically significant effects of the ‘user’ factor, with two degrees of freedom (DF), and the ‘hand’ factor, with one degree of freedom (DF). Statistically significant interaction between the factors were found in each sensor, as revealed by sums of squares (SS), mean squares (MS), and the F statistics with their associated probability limits (P) relative to the comparison given, as shown here below in Table 2.

Statistically significant two-way interactions are present in each of the sensors here and reflect the fact that users with different levels of training or task expertise use significantly different grip force control strategies. These are reflected by variations in the recordings from different sensors, as shown and discussed in detail in our previous work [23,24]. Effect sizes underlying significant variations, between individuals and hands, are given above in terms of differences in means and their standard errors (Figure 6a–c). To address the goal of this study here, i.e., reveal functionally distinct grip force co-variation patterns as a function of task expertise, correlation analyses using Pearson’s product moment were performed on paired sensor data, one by one, from each user and hand.

### 3.4. Correlation Analyses

Pearson’s product-moment correlation is expressed in terms of a coefficient *r*, shown in (1) here below, as a measure of the strength of a linear association between two variables indicating how far away all data points for *x* and *y* are from the line of best linear fit. The coefficient can take a range of values from +1 to −1. A value of zero indicates that there is no association between the two variables *x* and *y*. Values for ***r*** greater than zero indicates a positive correlation, values less than zero a negative correlation, where the value of one variable increases as the value of the other variable decreases. The product moment correlation expressed in terms of the coefficient (*r*) is calculated by dividing the covariance of two variables (*x, y*) by the product of their standard deviations. The form of the definition involves a product moment that is the mean or first moment of origin of the product of the mean-adjusted random variables:*r* = cov(*x,y*)/(stddev(*x*) × stdev(*y*))(2)

Significant positive or negative correlations, in terms of *r* and the associated probability limits p, between sensor activities in each of the three users in their dominant and/or non-dominant hands are shown in Table 3 here below.

The correlation analyses on grip force signals from the dominant right hand of the novice user (Table 3, top) reveal a total number of 18 positive and statistically significant correlations between sensor signals recorded from sensor 3 and signals from sensors 5 and 8, between signals recorded from 5 and signals from 8, 9, 10, and 11, between signals from sensor 6 and signals from 8, 9, and 10, between signals from sensor 7 and sensors 9, 10, and 11, between signals from sensor 8 and sensors 9, 10 and 11, between signals from sensor 9 and sensors 10 and 11, and between signals from sensor 10 and sensor 11. Statistically significant negative correlations were found between signals from sensors 7, 12, 9, and 11. No significant correlations were found between signals recorded from the non-dominant hand of the novice. The correlation analyses on grip force signals from the dominant right hand of the dominant-hand-trained user (Table 3, middle) reveal a total number of three positive and statistically significant correlations between sensor signals recorded from sensor 5 and signals from sensors 8 and 10, and between signals from sensor 8 and signals from sensor 10. No significant correlations were found between signals recorded from the non-dominant hand of the dominant-hand trained user. The correlation analyses on grip force signals from the dominant left hand of the expert user (Table 3, bottom) reveal a total number of two positive and statistically significant correlations between sensor signals recorded from sensor 5 and signals recorded from sensor 10, and between signals recorded from sensor 6 and signals from sensor 10. In the non-dominant right hand, 4 statistically significant positive correlations were found between signals from sensor 5 and signals from sensors 6 and 11, between signals from sensor 6 and signals from sensor 11, and between signals from sensor 8 and signals from sensor 11. One statistically significant negative correlation was found between signals from sensor 10 and signals from sensor 11. Graphic representations of the sensor correlations are shown in Figure 7a (novice user), b (trained user), and c (expert user). The correlation analyses reveal clearly that users with different levels of task training and expertise deploy employ different grip force strategies during the task for manual control of the robotic device handles. Based on prior knowledge relative to the putative roles of the different finger and hand regions reviewed in the introduction here above, a functional analysis of the sensor signal correlations found in the different hands of the three users is performed.

### 3.5. Functional Analysis

Different grip force control strategies relating to difference in task expertise are reflected here by significant differences in grip forces in the different finger and hand locations reflecting different prehensile synergies at the central control levels in the neural networks of the somatotopically organized brain map in S1, as explained here above. While these central control levels are not directly observable in the grip force profiles, the correlations found here are the directly observable consequence of these central differences. In subtle precision tasks, the fewer fingers are used, the greater is the grip control [4], and the largely redundant functional design of the human prehensile system [5,8] allows to achieve optimally parsimonious force deployment and coordination, or functionally synergy, of as few fingers as necessary.

Functional optimization is progressively achieved during task training, probably through a functional re-organization of co-variation patterns aimed at minimizing prehensile redundancy, as suggested here below in Figure 7, which shows the strategy charts for grip force deployment in the dominant and non-dominant hands of the three user types. From the functional charts in Figure 8, it becomes clear that the expert’s dominant hand strategy is reflected by parsimonious correlation of grip forces from hand and finger regions ensuring gross grip force deployment and control: the middle phalanxes of the ring and middle fingers and the base of the thumb. The same force correlation is found in the non-dominant hand of this expert, who is proficient with either hand in this specific task. As a consequence, the grip force strategy charts for his two hands are rather similar, with the exception that the non-dominant deploys additional force correlations with regions located in the palm of the hand. The strategy chart of the novice, in contrast, is characterized by a large number of functionally non-specific correlations between grip forces from almost all finger and hand regions of the dominant hand. These different levels of expertise of the three users here are reflected by other task parameters such as average session times and total number of incidents, including number of drops, misses, and tool-trajectory adjustments (cf. see Table 1).

## 4. Discussion

In our previous work [23,24], using a similar multi-digit precision grip force robotic task, we showed that grip force control in a highly proficient task expert, by comparison with absence of control in an absolute beginner with no task training at all, is reflected by statistically significant differences between finger and hand locations ensuring specific functional roles [3,4,5,6,7,8,9,10,11] for fine versus gross grip force control. Here, we provide step-by-step statistics which show that different levels of proficiency in performing the highly complex precision grip task are reflected by skill-specific, distinct co-variation patterns in locally produced grip force signals from the most task-relevant sensor locations in the middle, ring and small fingers and in the palm of the preferred hand. These co-variation patterns are presumed to reflect individual grip force strategies, governed by prehensile synergies at the central control levels in the neural networks of a somatotopically organized cortical brain map [13,14,15,16,17,18,19,20,21,22]. With grip forces being both locally and globally mapped in the somatosensory brain [21,22], it is shown here that variations in task-skill result in distinctive from-global-to-local grip force co-variation patterns across fingers and hands. Prehensile redundancy [5,8] appears minimized in the skillful performance of the expert, while the complete novice deploys digit forces nonspecifically for lack of skill in the precision grip task, as revealed by the multiple non-specific co-variation patterns involving almost all sensors in his dominant hand. A challenging aspect of multivariate signal analysis is to decipher the functional significance in a multitude of simultaneously generated signal(s). To a given, albeit limited, extent, this has been achieved here by the analyses here, which highlight skill specific co-variation patterns in individual grip force profiles in a complex task-user system. These patterns are task specific [2,3,4,5,6,7,8], and one of the specific characteristics of the task-user system here is that it offers limited possibility of hand movement. This has the advantage that there are only a limited number of control strategies a user can employ to optimally perform the task at hand; in other words, different highly proficient experts are bound to use similar control strategies on this system This conveys additional significance to the data from the single expert here, which can be assumed representative. There are only a few experts able to use this kind of system skillfully. On the other hand, further studies on a larger variety of trainees and novices with longer periods of task training are needed to understand the evolution of co-variation patterns in the grip force signals with time. Many more than ten sessions would be required to achieve this, given that it takes more than hundred hours of training to become a task expert on this kind of system. Also, the sensor gloves here do not provide haptic feed-back to the user. Multi-modal feed-back systems help achieve near optimal average grip forces for optimal surgical precision, especially in minimally invasive surgery, producing lesser general grip force deployment during interventions and shorter intervention times [23,24,25,27,28,29]. Real-time interventional monitoring of individual grip forces by wearable devices combined with direct sensory feedback appears the ultimate solution. While the central control process underlying grip style management is non-conscious [30], grip force excess can be made accessible to immediate awareness of the user during an intervention. Thus, multi-sensor real-time grip force sensing by wearable systems can directly help to prevent incidents or tissue damage during interventions when it includes a procedure for sending specific visual or auditory signals [28,29,30,31,32,33] to the operator or surgeon whenever his/her grip force exceeds a critical limit.

## 5. Conclusions

Correlation analyses of grip force signals from multiple locations in the hands of individuals with varying levels of expertise in a complex precision grip task reveals skill-specific differences in the functional organization of local grip forces. These differences are reflected by distinct co-variation patterns in data recorded from multiple sources of variation in the dominant and non dominant hands. The co-variation patterns in the dominant hand of a proficient precision grip task expert, in contrast to those shown in the dominant hand of a novice, are consistent with optimized coordination of specific local grip forces and minimized prehensile redundancy. This functional interpretation is consistent with the current state of the art in neuromaging suggesting a from global-to-local mapping of grip force representations in the human brain. This conclusion/ interpretation needs to be investigated further by testing how the non-specific co-variations in the local grip forces of task novices evolve with time in hundreds of training sessions, necessary to become a true expert in the precision grip task from this study here. Developing this research further should contribute significantly towards training individuals on highly complex precision grip systems on the grounds of objective performance criteria beyond the time of task completion.

## Figures and Tables

**Figure 1 bioengineering-07-00143-f001:**
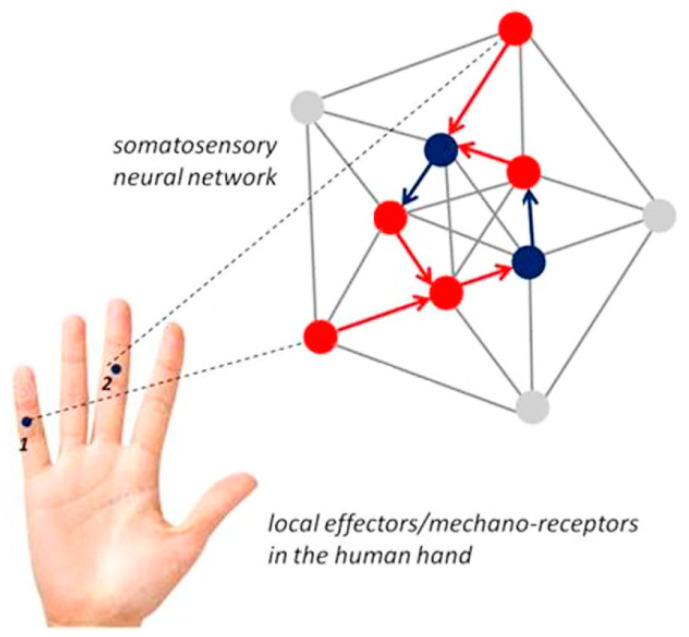
Illustrates how locally generated grip-force data may be encoded in a fixed-size neural network in the somatosensory brain [14,15,16,17,18,19,20,21,22]. Local effectors/mechanoreceptors on the middle phalanxes of the small (1) and the middle fingers (2) are indicated for illustration. The neural connectivity and global network representation are arbitrarily chosen here, for illustration only; single nodes in the network as shown may correspond to single neurons, or to a subpopulation of neurons with the same functional role. Red nodes may represent motor cortex (M) neurons, blue nodes may represent connecting visual neurons (V). Only one-way propagation is shown. It is assumed that different skill levels in grip control relate to different levels of representation in the brain producing functionally distinct grip force co-variation patterns across fingers and hands.

**Figure 2 bioengineering-07-00143-f002:**
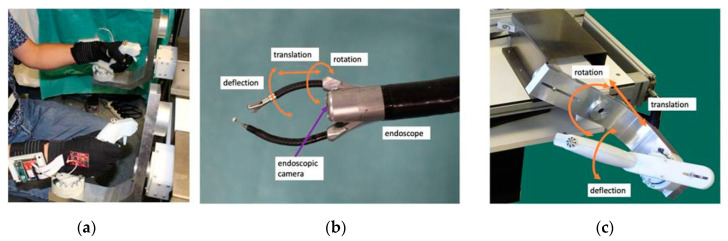
Expert wearing the sensor gloves while manipulating the robotic master/slave system (**a**). Master-slave control chart of the master/slave system (**b**). Direction and type of tool-tip and control movements (**c**).

**Figure 3 bioengineering-07-00143-f003:**
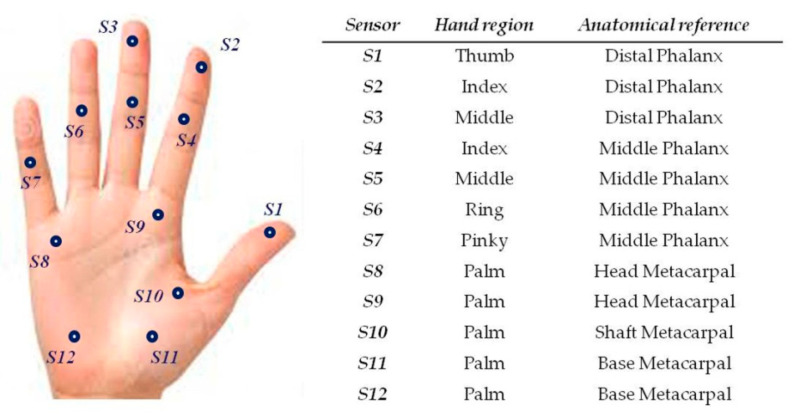
Sensor locations on the inner surface of the dominant or non-dominant hand.

**Figure 4 bioengineering-07-00143-f004:**
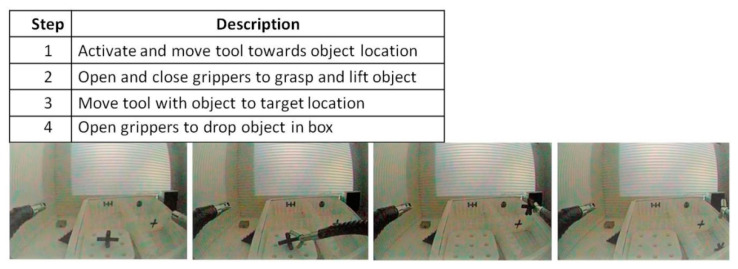
Description and visual illustration of the 4-step pick-and-drop task when performed with the right hand.

**Figure 5 bioengineering-07-00143-f005:**
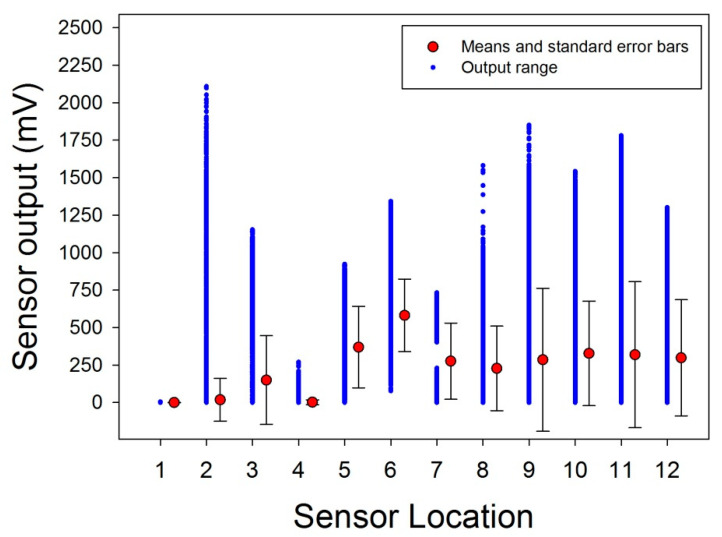
Means, standard errors, and output range between *minima* and *maxima* recorded from each of the twelve sensor locations across users and hands.

**Figure 6 bioengineering-07-00143-f006:**
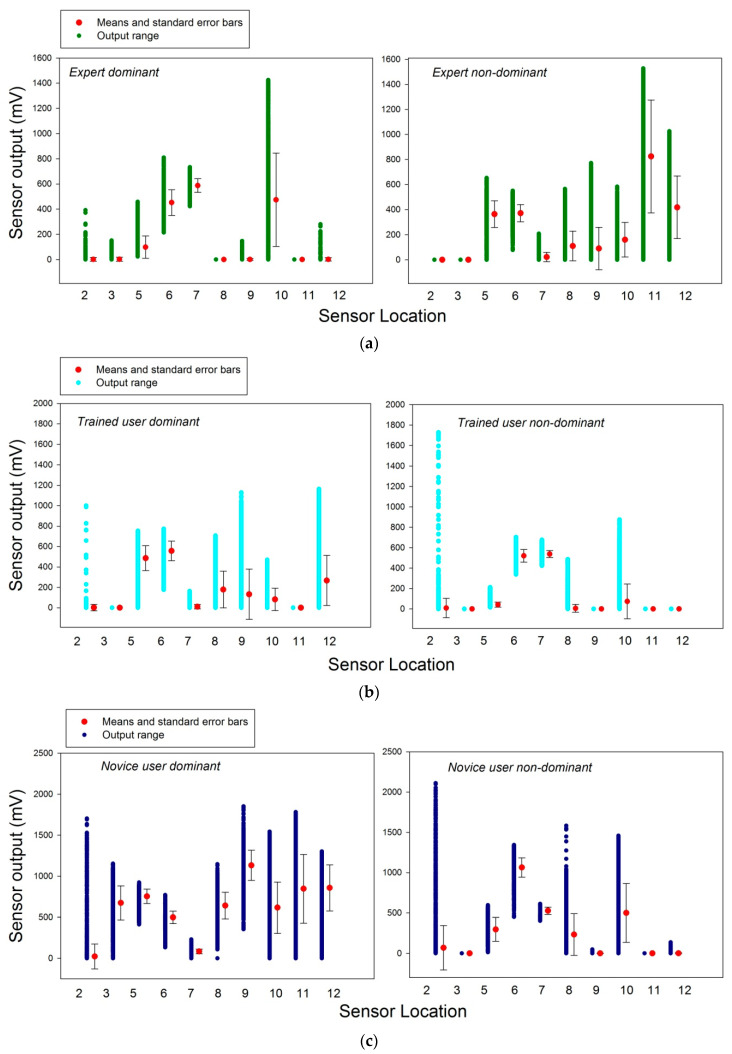
Means, standard errors, and output range between *minima* and *maxima* recorded from sensors that generated significant amount of signal output (>10% of signals recorded show non-zero activity) in the dominant and non-dominant hands of the expert user (**a**), the dominant-hand-trained user (**b**) and the novice (**c**).

**Figure 7 bioengineering-07-00143-f007:**
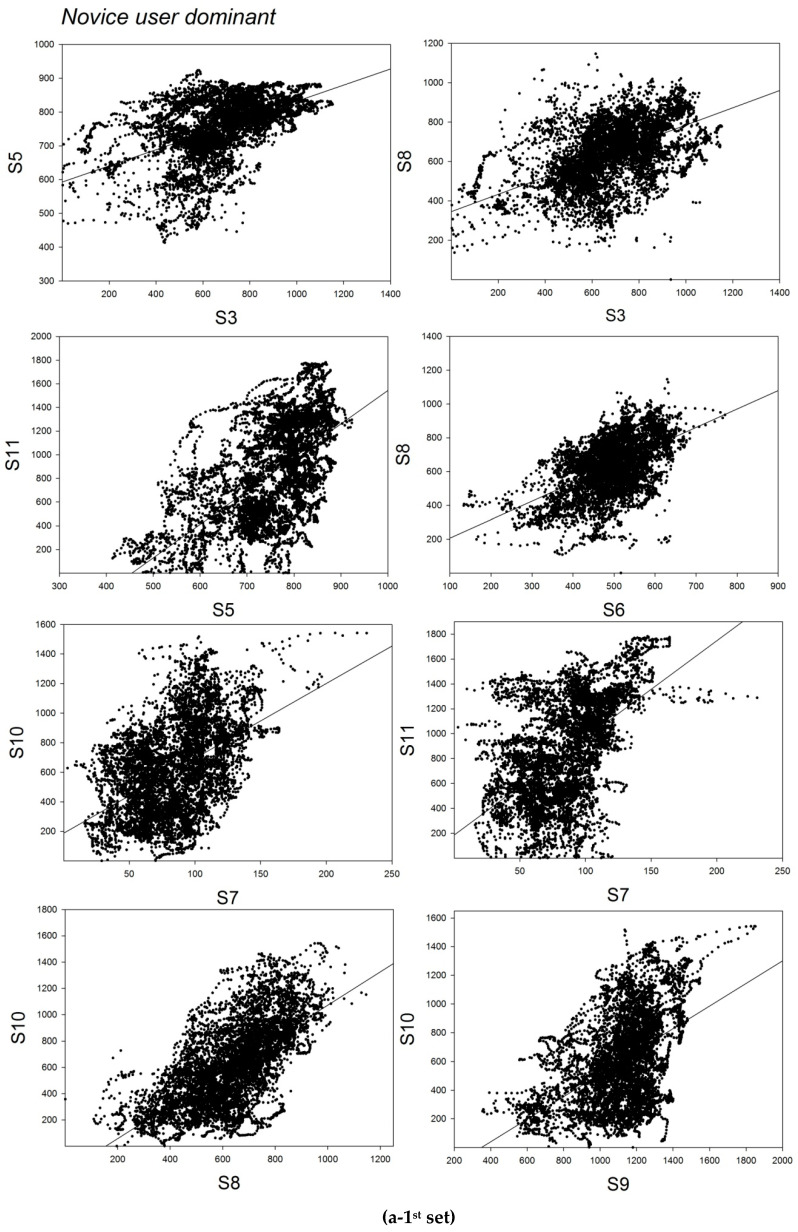
Graphic representations of significant positive or negative sensor correlations (Pearson’s product moment) between sensor signals in the dominant and/or non-dominant hands of the novice (**a**), the trained user (**b**), and the expert (**c**).

**Figure 8 bioengineering-07-00143-f008:**
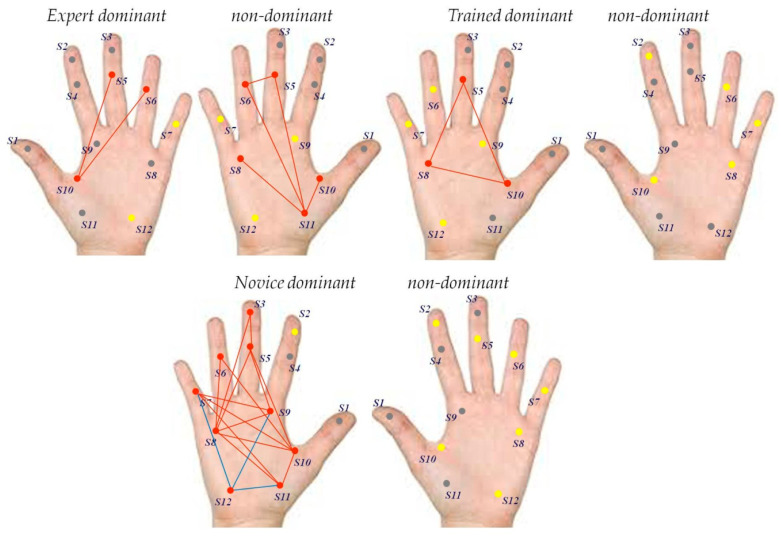
Functional strategy chart based on grip force signals and their positive/negative correlation, functional independence, or non-use in the precision grip task for the dominant and non-dominant hands of the expert (**top left**), the trained user (**top right**), and the expert (**bottom**). Non-activated grip force locations are highlighted as gray dots, independently active grip force locations as yellow, and significantly correlated locations as red dots. Red links between sensors indicate positive correlation, blue links indicate negative correlation.

**Table 1 bioengineering-07-00143-t001:** Average task completion time (in seconds) and cumulated number of incidents as a function of expertise levels and hand for ten repeated task sessions per individual.

Time	Incidents			
	dominant	non-dominant	dominant	non-dominant
**Novice**	15.42	12.99	20	28
**Trained**	11.90	13.53	6	8
**Expert**	8.88	10.19	3	0

**Table 2 bioengineering-07-00143-t002:** Results from 2-way ANOVA (general linear model) on individual sensor data as a function of user and hand.

	Source of Variation	DF	SS	MS	F	P
**S2**	User	2	16,773,987.886	8,386,993.943	427.929	<0.001
	Hand	1	2,964,227.230	2,964,227.230	151.244	<0.001
	User × Hand	2	3,755,368.124	1,877,684.062	95.805	<0.001
	Residual	38,271	750,074,008.984	19,599.018		
	Total	38,276	3,357,894,773.041	87,728.466		
**S3**	User	2	1,015,096,423.250	507,548,211.625	53,513.867	<0.001
	Hand	1	461,998,520.386	461,998,520.386	48,711.289	<0.001
	User × Hand	2	1,021,778,383.168	510,889,191.584	53,866.127	<0.001
	Residual	38,271	362,978,394.485	9484.424		
	Total	38,276	3,357,894,773.041	87,728.466		
**S5**	User	2	689,902,030.446	344,951,015.223	32,517.606	<0.001
	Hand	1	1,395,424,378.812	1,395,424,378.812	131,542.911	<0.001
	User × Hand	2	64,882,104.607	32,441,052.303	3058.131	<0.001
	Residual	38,271	405,983,767.489	10,608.131		
	Total	38,276	2,828,192,985.527	73889.460		
**S6**	User	2	894,178,539.095	447,089,269.547	58,203.191	<0.001
	Hand	1	380,801,423.473	380,801,423.473	49,573.674	<0.001
	User × Hand	2	695,805,856.940	347,902,928.470	45,290.867	<0.001
	Residual	38,271	293,979,646.508	7681.525		
	Total	38,276	2,235,427,686.091	58,402.855		
**S7**	User	2	8,131,916.849	4,065,958.425	2996.062	<0.001
	Hand	1	2,415,730,452.996	2,415,730,452.996	1,780,067.111	<0.001
	User × Hand	2	25,823,790.450	12,911,895.225	9514.323	<0.001
	Residual	38,271	51,937,603.692	1357.101		
	Total	38,276	2,482,530,139.867	64,858.662		
**S8**	User	2	1,182,330,765.802	591,165,382.901	24,162.125	<0.001
	Hand	1	488,723,290.475	488,723,290.475	19,975.110	<0.001
	User × Hand	2	162,302,567.526	81,151,283.763	3316.817	<0.001
	Residual	38,271	936,361,775.619	24,466.614		
	Total	38,276	3,051,959,544.861	79,735.593		
**S9**	User	2	2,331,382,489.784	1,165,691,244.892	54,035.820	<0.001
	Hand	1	1,875,323,478.829	1,875,323,478.829	86,930.946	<0.001
	User × Hand	2	2,332,903,972.891	1,166,451,986.446	54,071.085	<0.001
	Residual	38,271	825603632.161	21,572.565		
	Total	38,276	8,721,314,740.557	227,853.348		
**S10**	User	2	1,576,444,413.907	788,222,206.954	11,233.344	<0.001
	Hand	1	36,558,031.549	36,558,031.549	521.007	<0.001
	User × Hand	2	292,642,407.935	146,321,203.967	2085.296	<0.001
	Residual	38,271	26,854,02745.672	70,168.084		
	Total	38,276	4,653,698,608.118	121,582.679		
**S11**	User	2	1,479,234,266.725	739,617,133.363	10,404.163	<0.001
	Hand	1	2,866,636,795.456	2,866,636,795.456	40,324.860	<0.001
	User × Hand	2	1,479,234,266.725	739,617,133.363	10,404.163	<0.001
	Residual	38,271	2,720,630,817.886	71,088.574		
	Total	38,276	9,064,486,828.465	236,819.073		
**S12**	User	2	647,731,602.199	323,865,801.100	8786.483	<0.001
	Hand	1	2,434,904,432.621	2,434,904,432.621	66,058.985	<0.001
	User × Hand	2	647,217,679.584	323,608,839.792	8779.511	<0.001
	Residual	38,271	1,410,651,827.906	36,859.550		
	Total	38,276	5,733,258,645.998	149,787.299		

**Table 3 bioengineering-07-00143-t003:** Statistically significant correlations between sensor activities for each user.

NOVICE USER dominant hand
	**S5**	**S6**	**S7**	**S8**	**S9**	**S10**	**S11**	**S12**
**S3**	0.55*p* < 0.05			0.57*p* < 0.05				
**S5**				0.54*p* < 0.05	0.48*p* < 0.05	0.42*p* < 0.06	0.60*p* < 0.01	
**S6**				0.50*p* < 0.05	0.65*p* < 0.01	0.44*p* < 0.05		
**S7**					0.60*p* < 0.05	0.47*p* < 0.05	0.54*p* < 0.05	−0.45*p* < 0.05
**S8**					0.55*p* < 0.05	0.66*p* < 0.01	0.36*p* < 0.06	
**S9**						0.46*p* < 0.05	0.39*p* < 0.06	−0.35*p* < 0.06
**S10**							0.58*p* < 0.01	
**S11**								−0.46*p* < 0.05
TRAINED USER dominant hand
	**S5**	**S6**	**S7**	**S8**	**S9**	**S10**	**S11**	**S12**
**S5**				0.46*p* < 0.05		0.48*p* < 0.05		
**S8**						0.67*p* < 0.01		
EXPERT USER dominant and non-dominant hands
	**S5**	**S6**	**S7**	**S8**	**S9**	**S10**	**S11**	**S12**
**S5**		0.62*p* < 0.001				0.53*p* < 0.01	0.67*p* < 0.001	
**S6**						0.70*p* < 0.001	0.55*p* < 0.001	
**S8**							0.55*p* < 0.01	
**S10**							−0.69*p* < 0.01

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
