# Peer review of "Correlating Grip Force Signals from Multiple Sensors Highlights Prehensile Control Strategies in a Complex Task-User System"

_bioengineering, 2020, doi:10.3390/bioengineering7040143_

Round 1

Reviewer 1 Report

The present study uses a glove to monitor real-time grip force profiles in different experience users. This system has revealed skill-specific differences in individual grip force profiles across multiple sources of variation, functionally mapped to the somatosensory brain networks, ensuring grip force control and its evolution with control expertise which can be used for individual performance training in complex task-user systems are brought forward.

It is an interesting topic but I believe that several changes must be performed to improve the manuscript prior to its publication. These suggestions/comments are:

  • Show in a figure the four steps of the grip task. This will enhance the reader’s comprehension.
  • How is the level of expertise evaluated? Where they asked to do the task in the same exact way?
  • The gathered data was based on a sole repetition of the task or averaged after several repetitions. Please describe the processes regarding signal pre-processing method, feature extraction, identification of artifacts, etc…?
  • What is the software used to perform the Statistical analysis, please specify.
  • Graph 5 need to include the x-axis with the number of the sensors.
  • In Table 1 SS and MS are not defined. Despite minor data generation, sensor 1 and 4 should be included. From Table 1 there are significant differences in all sensors. It could be interesting to perform an AdHoc test to better understand between which groups are the differences.
  • Why are some plots in Figure 6 in different colors. Please make the different plots and layouts similar between users.
  • From reading the manuscript, it seems that the main finding is that the use of a glove with sensors to track movement and grip force can be applied to develop personalized training strategies. Agreeing with this comment, there seems to be limited information to extract this finding. Firstly, there is no control group. In other words, there is no comparison between subjects doing the task in multiple occasions without receiving the feedback from the glove system and those using the glove. By comparing both groups it will be more evident the impact of this system in reducing the training curve of complex tasks. Secondly, there is only one person per user experience group and the data gathered seems to involve only after performing one sole time the task instead of averaging over several repetitions. This could show the variability between movements. Unfortunately, there is no information regarding the number of repetitions performed per each user before data analysis. Thirdly, according to the available data it seems that the trained and the expert users indeed perform more specific movements but their patterns are different. This might suggest that two equally experienced users might perform the same task in different ways due to specific individual factors.
  • In the current conclusion the authors relate grip force signals with central (brain) control mechanisms. However, this hasn’t been discussed in the manuscript and if it has it is in a very superfluous manner. The reviewer suggests to rewrite to a more meaningful finding related with the discussed topics.

Author Response

Replies to Reviewer 1

Thank you for your valuable comments and suggestions. We have fully taken into account all of them by revising our paper thoroughly and accordingly as follows:

Q: Show in a figure the four steps of the grip task. This will enhance the reader’s comprehension

Reply: Done. We have added a new Figure 4 which shows images of the four critical task steps along with their description; please see page 7 of the revised manuscript. All changes to the manuscript are highlighted in blue for easy tracking.

Q: How is the level of expertise evaluated? Where they asked to do the task in the same exact way?

Reply: How the level of expertise in this specific task using the specific robotic system is evaluated is now clearly explained under 2.5 on pages 7 and 8, with a Table. Also, it is now explicitly stated therein that: “the robot-assisted precision grip task has limited degrees of freedom for hand and finger movement. Users inevitably have to perform the task in the same way generally speaking. The locally deployed grip forces will vary depending on how skilled a user has become in performing the task accurately.”

Q: The gathered data was based on a sole repetition of the task or averaged after several repetitions. Please describe the processes regarding signal pre-processing method, feature extraction, identification of artifacts, etc…?

Reply: We now clearly explain in the new paragraph 3.1 on page 8 that:

“All data analyses were performed on the original raw data from the ten or eleven repeated task sessions of each individual. A grip force signal from each sensor was acquired every 20 milliseconds in task time across sessions, therefore, a larger number of data from a given individual reflects longer task times. In the specific task here, different finger and hand locations are variably solicited during task execution. Significant differences as a function of task expertise can be expected, as shown in our previous work [21,22]. Here, prior to further statistical analysis, a descriptive evaluation was performed to establish which of the twelve sensors produced task-relevant output data in terms of signals of at least 50 mV in at least 10% of the data. The hypothesis in this study here is that optimal grip force control (task expertise) should be reflected by an optimized co-variation pattern in the finger forces of the expert. Testing for such requires comparing between signal distributions with roughly the same amount of data and variance, which holds for all comparisons that follow here. Significant outliers in terms of values exceeding twice the amount of variance in a given data distribution were not identified.”

Averaged data were only computed in the first step of the descriptive analyses, as explained in 3.2 on page 9, to compare for amount of signal content and variance in the data, which had to correspond to the criteria explained. The ANOVA and the correlation analyses described in 3.3 and 3.4 were preformed on the non-averaged (raw) sensor data that were eligible for statistical comparison and correlation analysis (equivalent amount of signal contents and variance), as is now clearly explained.

Q: What is the software used to perform the Statistical analysis, please specify.

Reply: We used SYSTAT for SIGMAPLOT_12, as is now clearly stated in 3.2 on page 11.

Q: Graph 5 need to include the x-axis with the number of the sensors.

Reply: Done.

Q: In Table 1 SS and MS are not defined. Despite minor data generation, sensor 1 and 4 should be included. From Table 1 there are significant differences in all sensors. It could be interesting to perform an AdHoc test to better understand between which groups are the differences.

Reply: SS (Sums of Squares) and MS (Mean Squares) are now explicitly given in the legend of Table 1. Why sensor data from locations 1 and 4 were not included in the ANOVA and correlation analyses is now clearly explained in 3.1, the new sub paragraph relative to data preprocessing on pages 8-9:

…”a descriptive evaluation was performed to establish which of the twelve sensors produced task-relevant output data in terms of signals of at least 50 mV in at least 10% of the data. The study goal here is to reveal functionally specific and potentially skill-dependent co-variation patterns in the finger forces of the expert by comparison with the non-experts.  All further statistical analyses towards this goal, such as analysis of variance on the raw data (general linear model), or between-sensor correlation statistics, require comparing between signal distributions with roughly the same amount of data and variance in the data. This holds for all comparisons that follow here after elimination of sensor locations 1 and 4.”

This topic of this article is focused on co-variation patterns in the individual grip force profiles. Expertise-specific differences between specific sensors have been extensively studied in our previous work, as is now more clearly explained on page 12:

“Statistically significant two-way interactions are present in each of the sensors here and reflect the fact that users with different levels of training or task expertise use significantly different grip force control strategies. These are reflected by variations in the recordings from different sensors, as shown and discussed in detail in our previous work [23,24]. Effect sizes underlying significant variations, between individuals and hands, are given above in terms of differences in means and their standard errors (Figure 6a,b,c). To address the goal of this study here, i.e. reveal functionally distinct grip force co-variation patterns as a function of task expertise, correlation analyses using Pearson’s product moment were performed on paired sensor data, one by one, from each user and hand.”

Q: Why are some plots in Figure 6 in different colors. Please make the different plots and layouts similar between users.

Reply: Good point. Done.

Q: From reading the manuscript, it seems that the main finding is that the use of a glove with sensors to track movement and grip force can be applied to develop personalized training strategies. Agreeing with this comment, there seems to be limited information to extract this finding. Firstly, there is no control group. In other words, there is no comparison between subjects doing the task in multiple occasions without receiving the feedback from the glove system and those using the glove. By comparing both groups it will be more evident the impact of this system in reducing the training curve of complex tasks. Secondly, there is only one person per user experience group and the data gathered seems to involve only after performing one sole time the task instead of averaging over several repetitions. This could show the variability between movements. Unfortunately, there is no information regarding the number of repetitions performed per each user before data analysis. Thirdly, according to the available data it seems that the trained and the expert users indeed perform more specific movements but their patterns are different. This might suggest that two equally experienced users might perform the same task in different ways due to specific individual factors.

Reply: We have recordings (raw data) for at least ten repeated sessions for the expert, eleven for the novice, and ten for the trained user. This is clearly stated on page 8, lines 313-321. The glove system does not provide haptic feed-back, it is used to measure grip force profiles as a function of skill, or task-specific proficiency, without performance feed-back. This is now made clearer in the new text on page 4, lines 170-171. Potential benefits of such feed-back are pointed out in the discussion section, lines 527+. As to variability in specific movements, such is excluded on this system, which has limited degrees of freedom, as clearly explained in the manuscript (cf. our reply above). Also, there are not many experts on this robotic system; as a consequence, the expert’s performance can beyond doubt be considered as a representative benchmark. The limitations of the conclusions regarding novices and moderately trained non-experts are now clearly pointed out (more data and research are needed) in the revised discussion on page 19:

 “…the task-user system here offers limited degrees of freedom for hand movements. This has the advantage that there are only a limited number of control strategies a user can employ to optimally perform the task at hand; in other words, different highly proficient experts are bound to use similar control strategies on this system This conveys additional significance to the data from the single expert here, which can be assumed representative. There are only a few experts able to use this kind of system skillfully. On the other hand, further studies on a larger variety of trainees and novices with longer periods of task training are needed to understand the evolution of co-variation patterns in the grip force signals with time. Many more than ten sessions would be required to achieve this, given that it takes days or weeks, not a few hours, of training to become a task expert on this kind of system.”

Q: In the current conclusion the authors relate grip force signals with central (brain) control mechanisms. However, this hasn’t been discussed in the manuscript and if it has it is in a very superfluous manner. The reviewer suggests to rewrite to a more meaningful finding related with the discussed topics.

Reply: The text parts relating the grip-force co-variation patterns to brain control have been thoroughly revised and extended, with additional references, to make this point clearer; see the revision in the introduction, pages 3-4:

“Current neuroanatomy of the brain circuitry that controls individual digit movements and grip forces generated during such movements displays a from-global-to-local functional organization. Neuroimaging studies have shown representations in motor cortex (M1) of individual digit movements, resulting either in largely overlapping neural activity patterns with specific voxels responding to all moving fingers at the same time [21], or in a strictly somatotopic organization (functional map) with locally ordered representations of individual finger digit movements (flexion), from thumb to pinky [22].Given that internal grip force representation appears both locally mapped and globally encoded in the neural networks of the somatosensory brain, the question addressed here in this study is whether variations in skill or expertise may result in distinctive from-global-to-local grip force co-variation patterns across fingers and hands. While there is digit redundancy for spontaneous grip force production in simple grasping tasks where no particular skill is required [5], such redundancy may be minimized in the skillful performance of a highly demanding precision task. In other words, while the complete novice deploys digit forces nonspecifically for lack of skill in a precision grip task, the skilled performance of the expert should display optimal correlation of specific finger forces.”

and the revised discussion, page 19:

“different levels of proficiency in performing the highly complex precision grip task are reflected by skill-specific, distinct co-variation patterns in locally produced grip force signals from the most task-relevant sensor locations in the middle, ring and small fingers and in the palm of the preferred hand. These co-variation patterns are presumed to reflect individual grip force strategies, governed by prehensile synergies at the central control levels in the neural networks of a somatotopically organized cortical brain map [13-22]. Grip forces being both locally and globally mapped in the somatosensory brain [21,22], it is shown here that variations in task-skill result in distinctive from-global-to-local grip force co-variation patterns across fingers and hands. Prehensile redundancy [5, 8] appears minimized in the skillful performance of the expert, while the complete novice deploys digit forces nonspecifically for lack of skill in the precision grip task, as revealed by the multiple non-specific co-variation patterns involving almost all sensors in his dominant hand.”

Reviewer 2 Report

The paper  present a study of draws from thousands ofsensor data, recorded from multiple spatial locations. The individual grip force profiles of a highly proficient left-hander (expert), a right-handed dominant hand-trained user, and a complete novice performing an image-guided and robot-assisted  precision task with the dominant or the non-dominant hand are analyzed using statistic methods.

It is need to be more explicit about the  controller

More information about the computer  (line 238)

Line 261 must be rewritten

Figure 5 - not include the OX and OY axes...

Table 2 must introduce the Grid

The References are not as Template

The Conclusion must be improved

Author Response

Replies to Reviewer 2

Thank you for your valuable comments and suggestions. We have fully taken into account all of them by revising our paper thoroughly and accordingly as follows:

Q: It is need to be more explicit about the  controller

Reply: Additional information concerning the control of the slave instrument by the master interface has been added. See lines 198+.

Q: More information about the computer  (line 238)

Reply: Done. See lines 198+.

Q: Line 261 must be rewritten

Reply: Done.

Q: Figure 5 - not include the OX and OY axes...

Reply: Fixed.

Q: Table 2 must introduce the Grid

Reply: Fixed.

Q: The References are not as Template

Reply: Fixed to the best of our ability.

Q: The Conclusion must be improved

Reply: Discussion and Conclusions were rewritten accordingly to provide a clear and consistent message about the findings and their implications, all changes made to the manuscript during the revisions are highlighted in blue for easy tracking.

Round 2

Reviewer 1 Report

The authors have improved the manuscript in comparison with the one submitted.